# Object-Level Data Augmentation for Deep Learning-Based Obstacle Detection in Railways

**Marten Franke \*, Vaishnavi Gopinath, Danijela Ristić-Durrant** 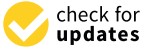 **and Kai Michels**

Institute of Automation, University of Bremen, Otto-Hahn-Allee, NW1, 28359 Bremen, Germany
* Correspondence: franke@iat.uni-bremen.de

**Abstract:** This paper presents a novel method for generation of synthetic images of obstacles on and near rail tracks over long-range distances. The main goal is to augment the dataset for autonomous obstacle detection (OD) in railways, by inclusion of synthetic images that reflect the specific need for long-range OD in rail transport. The presented method includes a novel deep learning (DL)-based rail track detection that enables context- and scale-aware obstacle-level data augmentation. The augmented dataset is used for retraining of a state-of-the-art CNN for object detection. The evaluation results demonstrate significant improvement of detection of distant objects by augmentation of training dataset with synthetic images.

**Keywords:** object detection; data augmentation; synthetic images; railway safety; long-range obstacle detection

## 1. Introduction

It is generally acknowledged that among land transport methods railways are comparatively safe. There are, nonetheless, each year many recorded collisions between trains and obstacles on or adjacent to the railway tracks, so rail operations could be made even more safe if the number of collisions could be reduced. Autonomous obstacle detection (OD) has gained significant interest in recent years as a recognized enabler of improved railway safety [1,2]. As a result of developments in artificial intelligence (AI), there has been a rapid expansion in research of deep learning (DL)-based methods and in particular of convolutional neural network (CNN)-based methods, for autonomous railway OD [3]. However, further progress in development and implementation of DL-based OD methods has been constrained by a lack of appropriate datasets that contain images including specific railway aspects, such as long-range obstacles on and near the rail track that are potentially hazardous for safe train operation.

In contrast to the road transport field, where a number of datasets such as KITTI [4] are available to support development and evaluation of DL-based methods for OD in road scenes, the rail transport field has, to the best of the authors' knowledge, only one relevant dataset, RailSem19 [5]. It is the first publicly available dataset explicitly for semantic railway scene understanding that contains both images acquired from the train's and tram's point of view and specific annotations for different railway relevant tasks, such as classification of trains, switches and buffer stops. However, RailSem19 does not include annotations of obstacles on the rail tracks. In the absence of relevant public datasets, the majority of AI-based methods for OD in railways are based on custom-made datasets, including images found on the Internet or images recorded during dedicated recordings in the railway environment. For example, the authors of [6] created the Railway Object Dataset that was used for development and evaluation of a CNN-based method for detection of different types of objects usually present in a railway shunting environment, such as: bullet train, railway straight, railway left, railway right, pedestrian, helmet, and spanner. As the dataset addressed the shunting environment, the assumed distances to obstacles were short-range.

However, the created dataset did not have object distance annotations, and actually did not consider object distances at all, which is also the case in the majority of published AI-based OD methods in railways.

To the best of the authors' knowledge, the only known published work that is based on a custom railway dataset with annotation of railway obstacles and associated distances is work included within the H2020 Shift2Rail projects SMART [7] and SMART2 [8]. The SMART railway dataset was created using data recordings in several static and dynamic field tests that were conducted on different railway sites in Serbia, according to permissions issued by the Serbian Railway. As illustrated in Figure 1a,b, during the static field tests, several cameras of different type, RGB, thermal, night vision and SWIR, were mounted on a static test stand viewing the rail tracks, with different obstacles located at different distances from the cameras, such as pedestrians that were imitated by the members of the SMART/SMART2 consortiums. In dynamic field tests, the cameras were integrated into a sensors' housing mounted on the front profile of an operational locomotive (Figure 1c,d), as described in [9] for the case of SMART obstacle detection system. Two different versions of the integrated sensors' housing were developed and tested during the successive projects, SMART and SMART2. As illustrated in Figure 1c, the SMART sensors' housing was horizontal and placed under the headlights of the locomotive, while the SMART2 sensors' housing was vertical and located in the middle of the locomotive frontal profile (Figure 1d).

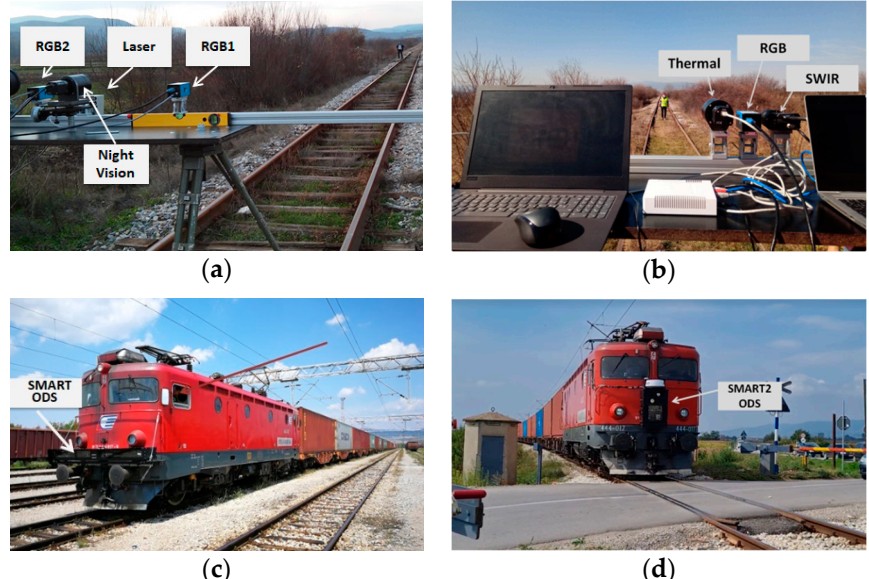

**Figure 1.** SMART and SMART2 field tests for dataset generation: Test-stand with the SMART (**a**) and SMART2 (**b**) sensors viewing the rail tracks and an object (human) on the rail tracks; SMART (**c**) and SMART2 (**d**) vision sensors for obstacle detection integrated into sensors' housing mounted on the frontal profile of a locomotive.

The dynamic field tests were performed with an in-service train of the operator Serbia Cargo. The test length was 120 km on the Serbian part of the pan-European corridor X. All the tests were performed in the period November 2017–June 2022, under different illumination and weather conditions. During the field tests, approximately 14 h of video were recorded by the SMART/SMART2 cameras. The majority of video images contain rail tracks without obstacles on them. In the images with obstacles, both static and moving obstacles are present, including humans, vehicles, and animals. Objects (obstacles) in the dataset were labeled with a number of parameters, including information about the class of the objects, the bounding box information of the object and the ground truth distance of the object to the cameras. In addition, objects in the dataset have associated instance segmentation masks, created manually by the online Hasty annotation tool [10]. These

instance segmentation masks were used by the authors in related works for annotating the images for training instance segmentation AI models [11]. For the data augmentation method presented in this paper, these objects' instance segmentation masks were used as described in Section 3.

The SMART dataset has supported development of the SMART/SMART2 on-board OD system, which is based on using state-of-the art bounding box DL-based models for object detection, that are originally trained with a public dataset such as COCO [12]. In the course of development of the SMART on-board OD system, different DL-based methods were used [13] as object detectors. In the recent advanced developments of the SMART2 on-board system, CenterNet [14], as one of the fastest and most reliable real-time networks for the object detection task, has been used. Although achieved results with the SMART vision-based OD system are satisfactory and they outperform other vision-based OD systems, particularly with respect to the distance detection range [15], there is still a need for further improvement. The reason for this is the insufficient amount of relevant training data. Namely, as said above, the number of images with "empty" rail tracks in the SMART dataset is much larger than the number of images with obstacles, which is understandable, bearing in mind that in the real-world operational environment during the data recording field tests, obstacles mainly appeared on level crossings and did not appear on large, secured areas. Furthermore, the viewed distances during the data recording field tests were limited by the topology of railway terrain. Because of this, there was a need to augment the SMART dataset, in particular to expand it with images of long-range distant obstacles. In order to overcome the problem of not having a sufficient amount of real-world data, in this paper, a novel method is proposed for generation of synthetic images to augment the SMART dataset and so enable re-training of the CenterNet object detector with the augmented dataset to improve its object detection performances.

The proposed novel augmentation scheme is a simple and efficient object-level data augmentation (ODA) method that improves the CNN model generalization for the railway obstacle detection domain. In order to achieve a context- and scale-aware data augmentation, the proposed data augmentation scheme includes a DL-based rail track detection as an essential step. The presented rail track detection method is a novel method based on using a state-of-the-art bounding box DL-based object detection, that detects parts of the rail tracks. For this purpose, the rail tracks in training images were annotated with multiple bounding boxes specifying rail tracks as objects consisting of multiple parts. Dealing with bounding box-based annotations and detection, the proposed rail track detection method is more efficient and faster than recent instance segmentation-based methods [16]. Additionally, as a DL-based method, it overcomes the shortcomings of established methods for rail track detection that are based on extraction of "hand-crafted" features, using traditional computer vision techniques [17]. Although focused on object-level data augmentation in the railway context, the proposed method can be used in any application where estimation of the object's distance is crucial for safety, such as autonomous driving [18].

The rest of the paper is organized as follows. In Section 2, an overview of related work focusing on generation of synthetic images and their use in the field of autonomous obstacle detection in railways is given. In Section 3, the proposed system for creation of synthetic images for training of a CNN model for obstacle detection in railways is presented. The experimental results are given in Section 4. The concluding remarks are given in the last section.

## 2. Related Work

DL-based methods, in particular CNN methods, are a very powerful AI tool that outperformed other techniques in object recognition tasks. However, the biggest shortcoming of this method is the immense amount of training data required, since the generation of a high-quality annotated dataset is an arduous and expensive process. Moreover, it is not always possible to obtain the needed data, because in many applications specific objects are to be considered and access to these objects is limited [19]. To tackle this problem, different techniques have been proposed for generation of synthetic data for object recognition tasks.

So-called cut–paste methods have been proven as simple methods that led to great results in many applications [20]. In principle, cut–paste techniques generate synthetic images by cutting the object from one image and pasting it on another image. In recent years, generative adversarial networks (GANs) have been introduced as a promising tool for synthetic image generation. A GAN is a model consisting of two networks: a generative network, which is trained to generate synthetic images, and a discriminative network, which is trained with images generated from the generative network and real images to classify images as real or synthetic [21].

So far, in the applications of OD in railways, cut–paste methods were not used for the generation of dataset images for training of an AI model for OD; rather, they were used to artificially inlay objects as possible obstacles on the images of rail tracks, for the purpose of generating test images for proposed OD methods. For example, due to the limited availability of real-world videos in which obstacles appear in front of trains, the methods presented in [22,23] were evaluated using modification of the videos, that were freely available on the Internet. The original high-resolution videos, recorded from the train driver's perspective, were modified by digitally adding stationary objects in the rail track regions in the video frames using appropriate software. Although the digitally added obstacles looked quite unrealistic, the synthetic images were satisfactory for evaluation of the proposed obstacle detection methods, representing a Windows-based systematic search for objects along the detected rail tracks.

Synthetic images were also used for evaluation of the proposed method in [24]. The proposed method was an unsupervised DL method for anomaly detection on the rail tracks. Due to the lack of a railway-driven dataset describing real-world obstacles over rails, the custom test dataset was generated using Gaussian–Poisson GAN (GP-GAN) [25], which is a generative model capable of blending images with high resolution, and generating realistic blends to inlay obstacles synthetically on images of rail tracks from the RailSem19 dataset. A total of 19 background images with rail tracks free of obstacles were modified by inlying obstacles using the GP-GAN. Each empty background image had an average of 10 inlaid counterparts. The obstacles were of typical classes such as "humans" and "animals". However, no details were given on the obstacles' inlying positions on the background images and, consequently, no references to the obstacles' distances were given.

Synthetic images for possible use as test images, as well as training images, for object detection models were created using the method presented in [26]. This method included the generation of synthetic images of a railway intruding object (potential obstacles) based on an improved conditional deep convolutional GAN (C-DCGAN), in order to overcome the absence of potential obstacles in the image datasets gathered from real railway operations. To investigate the authenticity and quality of the generated intruding objects, the generator, multi-scale discriminators, and a novel loss function of the improved C-DCGAN model were constructed. An intruding-object scale estimation algorithm based on a rail track gauge constant was used to generate and add synthetic intruding objects with accurate scaling into railway scene images taken from surveillance videos along high-speed rail lines. The object samples (classes: human, horse, cow and sheep), which were used for generating these synthetic objects, were extracted from semantic object labels of publicly available datasets. The experimental results on the created dataset demonstrated that the synthetic object images were of a high quality, diversity, and scale accuracy. However, the presented method augments the dataset with modified background images from static surveillance cameras, which limits its application for long-range on-board obstacle detection.

In general, published GAN-based methods to-date demonstrated GANs as good at generating natural images from a particular distribution, but weak in capturing the high-frequency image details such as edges [25]. In addition, DL-based methods are computationally expensive. Because of these reasons, although promising, GANs were not used for synthetic image creation in the work presented in this paper. Rather, a simple and efficient cut–paste-like method that includes novel DL-based rail track detection, which

supports scale-based data augmentation, was used to create the images for retraining of the CenterNet model for obstacle detection in railways, as explained in the following sections.

## 3. Proposed Method

The illustration of the pipeline to create synthetic images is shown in Figure 2, and a short explanation of individual processing steps is given in the following.

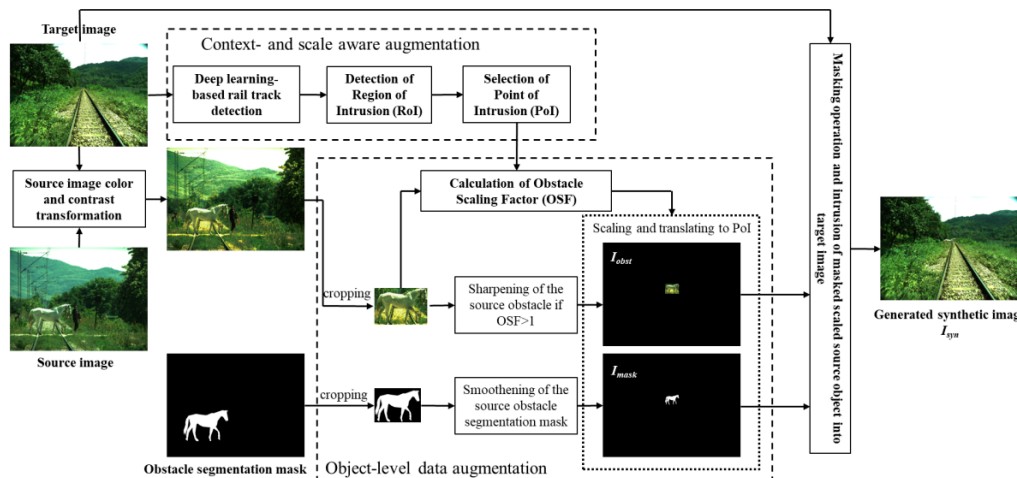

**Figure 2.** Block-diagram of proposed system for generation of synthetic image for obstacle detection in railways.

As illustrated in Figure 2, the presented ODA method takes three inputs from the SMART railway dataset explained in Section 1: a target image of the rail tracks without obstacles, a source image with an obstacle on the rail tracks and an offline-created instance segmentation mask of the obstacle from the source image. As described in Section 1, the used instance segmentation masks were part of the dataset and they were created manually. In principle, the method of creating instance segmentation masks could be automatized by integration of AI-based instance segmentation of source images. For the novel data augmentation method presented in this paper, it was sufficient to use offline-created objects' instance segmentation masks, as the associated objects were manually randomly selected. The main selection criteria were properties of the objects (such as size, class, and the location in the scene) that were illustrative for the presented data augmentation method. Figure 2 illustrates an example of source image, with a horse passing over the level crossing while a train is approaching it. The image was recorded in a SMART field test, with SMART cameras mounted on the train running on a test route approved by Serbian Railway. The crossing of the horse was a part of the field test scenario, and it was conducted in controlled conditions with the support of the local equestrian club near the town Niš, in Serbia.

The proposed method follows the idea of cut–paste techniques, and it creates a synthetic image by cutting the obstacle from the source image and inserting it into the target image using the segmentation mask. In order to achieve a realistic look in the generated synthetic image, the inserted obstacle should match with the background of the target image. For this purpose, at first, a color and contrast transformation operation is applied to the source image, considering the target image as a reference to reduce the color/illumination differences between foreground and background in the final synthetic image. In the presented system, a combination of two methods is proposed, histogram matching [27] and LAB color space-based blending [28]. Histogram matching modifies the contrast level of the source image according to the contrast level of the target image. Histogram matching is achieved by equalizing the histograms of the source and the target images and then by mapping the equalized pixel intensities of target to source image. LAB color space-based blending is a fast color transfer algorithm between two images represented in the LAB color space, where the L channel stands for luminosity/illumination/intensity, and the A and B

channels represent the green–red and blue–yellow color components, respectively. After transformation of both images, target and source image, from RGB to LAB color space, the three channels are split and the means and standard deviations of the three channels are calculated for both images. The objective of transferring the colors of the target image to the source image is achieved by an appropriate combination of means and standard deviations of three channels of both images. For the final transformation of the color and contrast of the source image, the outcomes from the above mentioned two methods are combined in so-called Alpha blending [29], based on the following Formula:

$$P_\alpha = P_{lab}(1 - \alpha) + P_{hist}(\alpha) \tag{1}$$

where $P_{hist}$ is the pixel intensity of the source image after histogram matching, $P_{lab}$ is the pixel intensity of the source image after LAB color-matching method, and $\alpha$ is a parameter $0 \leq \alpha \leq 1$, that is in the presented system considered to be equal to 0.3, based on heuristic evaluation of results. $P_\alpha$ is the pixel intensity of the pre-processed source image with the color/contrast transformed according to the color/contrast of the target image. The pre-processed source image is the input to the processing steps of the presented ODA, as illustrated in Figure 2.

### 3.1. Object-Level Data Augmentation

Firstly, the pre-processed source image containing the source obstacle and the image with obstacle instance segmentation mask are cropped to the size of the obstacle's bounding box. The cropped source obstacle is then sharpened by applying a sharpening filter if the obstacle scaling factor (OSF) is larger than 1, to retain the sharp object edges, even in the case that the obstacle shall be upscaled for the purpose of so-called true-to-scale insertion into the target image. In parallel, the cropped obstacle segmentation mask is processed by applying appropriate morphological operations with the goal to smoothen the edges of the binary segmentation mask. The processed, cropped source image and cropped segmentation masks are placed, respectively, into two images of the sizes of the target image, whose pixels all have zero values. These two images are denoted as $I_{obst}$ and $I_{mask}$. The points of insertion of the cropped images into the images $I_{obst}$ and $I_{mask}$ are referred to as points of intrusion (PoI). The PoI is defined as the image coordinates of the point in the target image where the bottom right corner of the source obstacle will be upon its insertion into the target image. The PoI and the OSF are calculated in the processing pipeline, named context- and scale aware augmentation, that consists of several processing steps as explained in the following.

### 3.2. Rail Track Detection

As said in Section 1, the presented ODA integrates autonomous rail track detection using DL-based object detection. More precisely, CenterNet that is used for OD in the SMART/SMART2 system is also used for the detection of rail tracks. For the purpose of training CenterNet to detect rail track parts, 234 images with rail tracks in the SMART dataset were annotated so that the visible rail tracks were annotated with multiple bounding boxes. The number of bounding boxes used for annotation depended on the length of the rail tracks in the image. An example of annotation of straight rail tracks with six bounding boxes is given in Figure 3a.

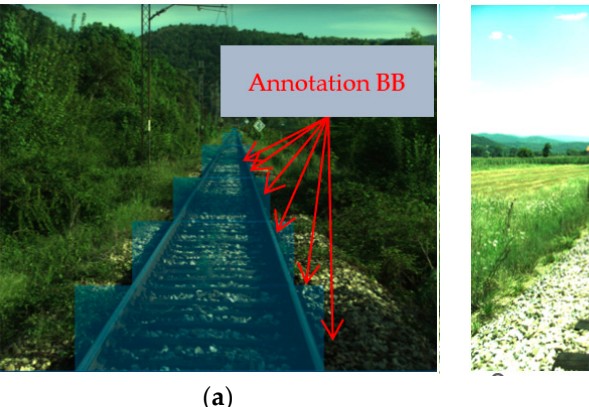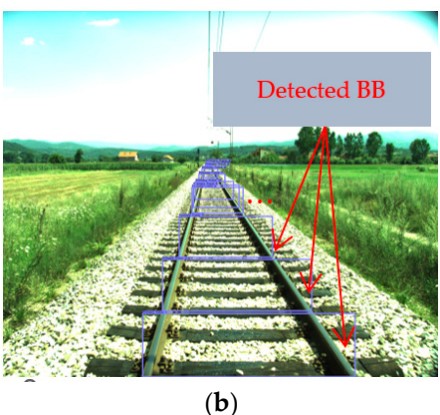

(**a**)　　　　　　　　　　　　　　　(**b**)

**Figure 3.** (**a**) Annotation of rail tracks with multiple bounding boxes (BB) along the visible length of the rail track. (**b**) CenterNet detection result of detected rail track parts as marked with purple bounding boxes.

During the annotation process, when selecting the parameters of the annotation bounding boxes, the following criteria were applied:

- The bottom left and the bottom right corners of each bounding box have to lie exactly on the outer edge of the left and right rail track, respectively;
- The ratios of height and width of the individual bounding boxes are approximately constant.

The 234 original images were selected so that the annotated dataset contained diverse data: images with different backgrounds as captured in different weather and illumination conditions, and images with different structure of the rail tracks terrain. The CenterNet model, originally trained on the COCO dataset, was retrained with annotated rail track images using the following hyperparameters: learning rate: $2.5 \times 10^{-4}$, number of epochs: 230, batch size: 4, learning rate drop at epoch 180 and epoch 210, input resolution: 1024.

An example of the CenterNet model detection is given in Figure 3b. As can be seen, the corner points of the bottom sides of all bounding boxes lie very precisely on the rail tracks, so that linear regression lines obtained by curve-fitting through these points represent the rail tracks reliably.

The left/right regression line is formed starting from the left/right bottom corner of the lowest bounding box in the image ($y_{max}$ coordinate) to the left/right upper corner of the uppermost bounding box ($y_{min}$ coordinate). Firstly, the average $x$- and $y$-coordinates of bottom corners of all bounding boxes, $\bar{x}$ and $\bar{y}$, and the standard deviations, $\sigma_x$ and $\sigma_y$, are determined. The correlation factor $r$ is assumed to be 1 for the sake of simplicity. The slope of the regression line $m$ can be determined using the equation:

$$m = r \cdot \frac{\sigma_x}{\sigma_y} \qquad (2)$$

While the coordinates $y_{max}$ and $y_{min}$ for both regression lines are known through the coordinates of the bounding boxes obtained as the outcome of the CenterNet-based rail track detection, corresponding $x$-coordinates have to be calculated from the Equations (3) and (4) for the left regression line as well as (5) and (6) for the right regression line:

$$x_{L,max} = \bar{x} + \frac{\bar{y} - y_{min}}{m} \qquad (3)$$

$$x_{L,min} = \bar{x} - \frac{y_{max} - \bar{y}}{m} \qquad (4)$$

$$x_{R,max} = \bar{x} + \frac{y_{max} - \bar{y}}{m} \qquad (5)$$

$$x_{R,min} = \overline{x} + \frac{\overline{y} - y_{max}}{m} \tag{6}$$

An example of calculated regression lines drawn onto the original image of rail tracks is given in Figure 4a. As can be seen, the calculated regression lines map the courses of the left and right rail track accurately so that they can be used for the definition of a region of intrusion (RoI) and the obstacle scaling factor (OSF), as explained in the following.

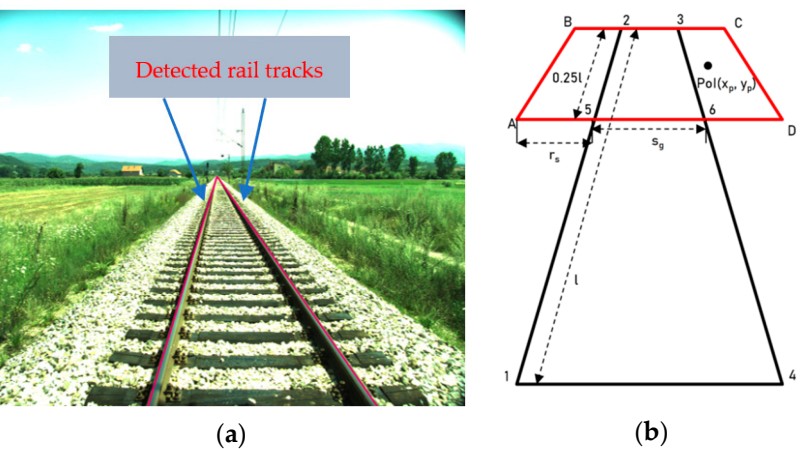

(a)                                      (b)

**Figure 4.** (**a**) Detected rail tracks represented by red lines overlaying the original image. (**b**) Illustration of region of intrusion (RoI) definition (red trapezoid).

### 3.3. Definition of Region of Intrusion (RoI)

The goal of the presented ODA method is to create synthetic images to augment the existing SMART dataset with images of long-distance obstacles that are hazardous for safe rail transport. To cover the long-distance range, the length of the RoI in the target image is calculated as the farthest 25% of the total length of detected rail tracks. As illustrated in Figure 4b, once the regression lines are calculated to represent left/right rail tracks, the start and the end points of both lines are detected (points 1 and 2 for the left rail track, i.e., points 3 and 4 for the right track). If the length between the start and end points of the detected rail tracks is denoted as $l$, the length of RoI is calculated as $0.25l$. For all rail track images in the SMART dataset, this definition of the RoI length corresponded to real-world distances beyond 200 m.

In order to cover the so-called clearance region, the width of the RoI is calculated based on the standard real-world size of the rolling stock and its width outside the rail tracks. According to [30], the standard width of the rolling stock's protrusion outside the rail tracks is $Rs = 0.895$ m. Considering the standard-gauge railway with a track gauge of $Sg = 1.435$ m, and considering the projective transformation by camera imaging:

$$r_s = s_g \cdot \frac{R_s}{S_g} \tag{7}$$

where $s_g$ is the distance between the corresponding points on the detected left and the right track (track gauge) in pixels, the pixel width $r_s$ of the RoI outside the rail tracks on both the sides of the rail track can be calculated. In this way, the RoI trapezoid A-B-C-D, is defined as illustrated in Figure 4b.

### 3.4. Calculation of Obstacle Scaling Factor (OSF)

In order to define a so-called point of intrusion (PoI) within the RoI, that will define the point in the target image to be aligned with the bottom right corner of the source obstacle bounding box upon its inserting into the target image, a random point generator was applied which was able to generate a random PoI with image coordinates $(x_p, y_p)$. Multiple PoIs are generated for every source obstacle and for every target image for the

purpose of synthetic obstacle intrusion. These PoIs are uniformly distributed across the entire RoI of the target image.

In order to ensure true-to-scale insertion of the source obstacle into the target image, an OSF has to be calculated considering the pixel width of the rail tracks at the $y_p$-coordinate of PoI as follows:

$$\text{OSF} = \frac{h_p}{h_{\text{source}}} \tag{8}$$

where $h_{\text{source}}$ is the obstacle height in source image (pixels) and $h_p$ is the obstacle height in pixels in the final synthetic target image $I_{\text{syn}}$. $h_p$ is calculated based on the width of the detected rail tracks in the target image at the $y$-coordinate of the PoI as:

$$h_p = s_{gp} \cdot \frac{S_g}{H} \tag{9}$$

where $s_{gp}$ is the distance between the corresponding points on the detected left and the right track (track gauge) in pixels at $y_p$-coordinate, and $H$ is the averaged real-world height of the objects from a particular class.

### 3.5. Insertion of Source Obstacle into the Target Image

Once the images $I_{\text{obst}}$ and $I_{\text{mask}}$ are created, as explained in Section 3.1, by inclusion of, respectively, the scaled pre-processed source obstacle and scaled obstacle segmentation mask at PoI, while all other pixels are of 0 values, the masking operation is performed to insert the scaled obstacle into target image $I_{\text{target}}$ as follows:

$$I_{\text{syn}}(x, y) = \begin{cases} I_{\text{obst}}(x, y) & \text{if } I_{\text{mask}}(x, y) \neq 0 \\ I_{\text{target}}(x, y) & \text{otherwise} \end{cases} \tag{10}$$

## 4. Evaluation

Using the presented ODA method, a significant number of synthetic images were generated. For every obstacle from selected source images, a set of ten unique target images from SMART dataset was used. In each target image, the same object was inserted at five different positions upon appropriate scaling so that five synthetic images were created per pair "source object/target image". This procedure led to 50 different synthetic images per one source obstacle. All created images were finally filtered by subjective visual inspection so that all synthetic images of poor quality with a non-realistic appearance were discarded. Finally, 2026 synthetic images were used for re-training of the CenterNet object detection model.

In order to demonstrate the impact of augmentation of the current SMART dataset with the generated synthetic images on the performance of DL-based object detection, two different CenterNet models were trained:

- Model 1—the object detection model trained exclusively with the annotated real-world images of the original training dataset;
- Model 2—the object detection model trained with the augmented training dataset which was a mixture of real-world and synthetic images.

When creating the original real-world dataset of 1005 images in total, 80% were assigned to the original training dataset, and 10% of the images were used for each of validation dataset and test dataset. The assignment of the individual real-world images to the training, validation or test dataset was done randomly. The same validation and test datasets were used for Model 1 and Model 2, because the evaluation of the training results should be based on the same real-world images. The training dataset for Model 2 was augmented with respect to training dataset of Model 1 with synthetic images. The sizes of training, validation and test datasets for Model 1 and Model 2 are given in Table 1 below:

**Table 1.** Number of images for training, validation and test datasets.

|         | Training Dataset | Validation Dataset | Test Dataset | Total |
|---------|------------------|--------------------|--------------|-------|
| Model 1 | 804              | 100                | 101          | 1005  |
| Model 2 | 2830             | 100                | 101          | 3031  |

Although the number of synthetic images significantly exceeded the number of real-world images, the total number of objects in real-world images, 4088, was much larger than the total number of objects in synthetic images, 2026. This is because there was only one synthetically inserted object per synthetic image while most of the original real-world images contained multiple objects.

To ensure comparability of the two models, both models were trained with the same hyperparameters: learning rate: $1.5 \times 10^{-4}$, number of epochs: 140, batch size: 4, learning rate drop at epoch 90 and epoch 120, input resolution: 1024.

The average precision (AP) is used as the decisive metric for the evaluation of the models. As shown in Table 2, the mean average precision (mAP) across all intersection over union (IoU) values increased absolutely by 3.8% and thus experienced a significant improvement due to the augmentation of the original training dataset with synthetic images.

**Table 2.** Evaluation results.

|                              | Model 1 (Original Dataset) | Model 2 (Augmented Dataset) |
|------------------------------|----------------------------|-----------------------------|
| mAP @ IoU = [0.5:0.05:0.95]  | 0.486                      | 0.524                       |
| AP @ IoU = 0.5               | 0.691                      | 0.776                       |
| AP @ IoU = 0.75              | 0.527                      | 0.562                       |
| AP @ distant                 | 0.305                      | 0.402                       |
| AP @ medium                  | 0.473                      | 0.509                       |
| AP @ near                    | 0.645                      | 0.633                       |

A qualitative analysis of the predictions on the test dataset showed that Model 2 often produced better predictions. Object detection using Model 1 resulted in more false positives (FP), and also false negatives (FN) occurred more frequently. Thus, both sensitivity and specificity were increased for Model 2. In Figure 5, an example of this improvement of sensitivity and specificity is shown.

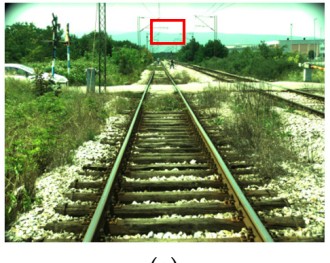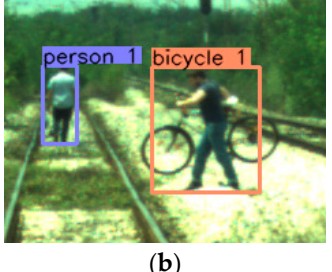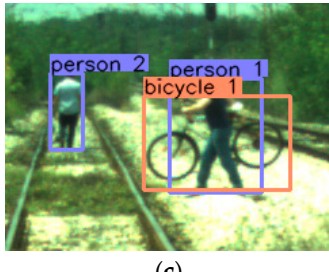

(**a**)  (**b**)  (**c**)

**Figure 5.** Example of improved sensitivity and specificity. (**a**) Original image with marked region containing the considered objects (red rectangular). (**b**) Magnified predictions of Model 1 including one FP and two FN predictions. (**c**) Magnified predictions of Model 2 with perfect predictions.

Furthermore, a more detailed evaluation of the predictions led to the result that the bounding boxes of the recognized objects of Model 2 are repeatedly more accurate than those of Model 1 (Figure 6). This led to increased IoU values, which can have a considerable influence on the mAP, and is therefore another reason for the improvement of the mAP presented in Table 2.

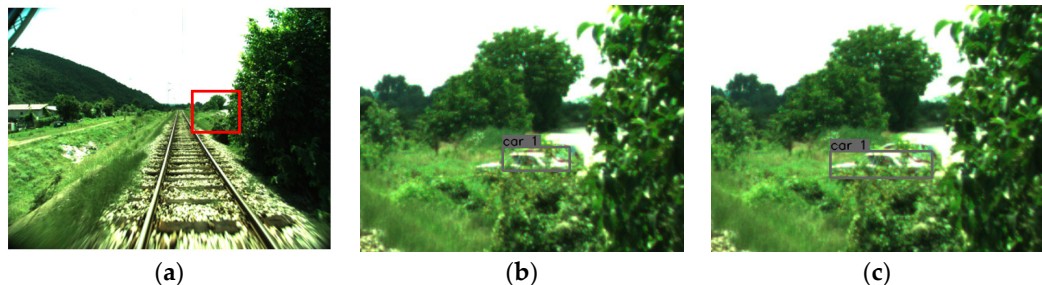

<div align="center">(<b>a</b>)        (<b>b</b>)        (<b>c</b>)</div>

**Figure 6.** Example of improved object bounding box coordinates. (**a**) Original image with marked region containing the considered object (red rectangular). (**b**) Magnified predictions of Model 1. (**c**) Magnified predictions of Model 2.

Even though the more precise detection of bounding boxes may not seem particularly relevant at first, it is of great importance for the distance estimation based on the bounding box sizes, as it is carried out in the SMART/SMART2 system using the artificial neural network DisNet [13]. Accordingly, more precise bounding boxes have a direct influence on the accuracy of distance estimation of objects. For example, the ground truth distance between the camera mounted on the train and detected object (car) in Figure 6 was 93 m. The estimated distance based on the imprecise car bounding box resulting from Model 1 was 125 m, while the estimated distance based on the precise bounding box resulting from Model 2 was 104 m.

Overall, the newly trained Model 2 was able to correctly detect 221 of the 252 objects present in the test dataset. In all these cases, the IoU was above 0.5 and the object was classified as true positive (TP). Accordingly, 31 FN occurred in the test dataset. Furthermore, 35 FP occurred, resulting in a improved precision of 0.863 and a recall of 0.877. Thus, the harmonic mean (F1 score) for this model is 0.870, as can be seen in the following Table 3.

**Table 3.** Precision, recall and F1 score.

| Metric | Model 1 | Model 2 |
|:---:|:---:|:---:|
| Precision | 0.824 | 0.863 |
| Recall | 0.857 | 0.877 |
| F1 score | 0.840 | 0.870 |

In the railway application, bearing in mind the long braking distances of a train, early obstacle detection in long-range distances is of great importance. Based on the assumption that small objects in SMART test dataset images correspond to distant objects (small with respect to the number of pixels they are covering in the image), the evaluation was performed on how the object detection model re-trained on the augmented dataset, with respect to the ability to detect distant objects. First, all objects were assigned to one of the three categories, "near", "medium" and "distant", so that three equally distributed classes were created. The threshold between "near" and "medium" category was $135^2$ pixels per object, which corresponds to a distance of approximately 100 m. The second threshold between "medium" and "distant" categories was $46^2$ pixels per object, which corresponds to a distance of approximately 250 m. As Table 2 shows, training with the augmented dataset resulted in an improved AP for the "distant" and the "medium" objects. While in the categories "medium" and "near", only a minimal improvement or worsening of the AP value emerges, in the category "distant" a significant increase of the AP by 9.7% has been achieved. Due to its high importance for obstacle detection in railways, this improvement of detection of distant objects by augmentation of training dataset with synthetic images was one of the primary goals of the presented work. The decrease of 0.012 in the AP value for near objects is probably due to the fact that most of the objects in the synthetic images were inserted in the medium or distant distance range. This resulted in

less focus on the near objects during the training of Model 2 than during the training of Model 1, and thus a lower AP value was obtained in this distance range.

## 5. Conclusions

In this paper, a novel, effective and fast method for creation of synthetic images for DL-based obstacle detection in railways is presented. The method is an end-to-end technique that selects an input source image containing an obstacle from the railway dataset and detects the rail tracks in selected target background image, based on which the long-distance region of obstacle intrusion in the target image is calculated. Detected rail tracks are also used to calculate the obstacle scaling factor, which enables scale-aware insertion of the source obstacle into the target image. The presented rail track detection method proves a novel use of the CenterNet object detector for detecting of objects that are annotated with multiple bounding boxes in training images. Experimental results demonstrate improvement in DL-based obstacle detection over long-range distances on the straight rail tracks due to augmentation of the training dataset. A limitation of the presented method is that it functions well only for the straight rail tracks. Hence, in the authors' future work the method will be modified to include curves on the rail tracks as well.

**Author Contributions:** Conceptualization, M.F. and D.R.-D.; validation, M.F.; writing—original draft preparation, V.G.; writing—review and editing, M.F. and D.R.-D.; visualization, M.F.; supervision, D.R.-D. and K.M.; project administration, D.R.-D. All authors have read and agreed to the published version of the manuscript.

**Funding:** This research received funding from the Shift2Rail Joint Undertaking under the European Union's Horizon 2020 research and innovation program under Grant No. 881784.

**Institutional Review Board Statement:** Not applicable.

**Informed Consent Statement:** Not applicable.

**Data Availability Statement:** Not applicable.

**Conflicts of Interest:** The authors declare no conflict of interest.

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
