# Peer review of "Object-Level Data Augmentation for Deep Learning-Based Obstacle Detection in Railways"

_applsci, doi:10.3390/app122010625_

Round 1

Reviewer 1 Report

The authors address the relevant problem of obstacles detection on railway tracks. The proposed approach for data augmentation is relatively sophisticated, however it relies on significant simplifying assumptions. The importance of such data augmentation is clearly demonstrated by the authors, given the small amount of currently available data.

I do have some concerns though, that prevent me from recommending the paper in its present form:

1. How were the obstacles masks generated, manually or with an object segmentation algorithm? In the latter, how can one automate the procedure so that it is reliable without a human intervention?

2. The way the dataset has been created is unclear: where do these 804 images come from? This is close but not equal to 993 x 80%. And, similarly, 109 images in the validation set are not 993 x 10%. Moreover, how were these images chosen, randomly?

3. How have the validation and test sets been used? Are the results in Table 1 from one of the two? From the text it is hinted that these are calculated on the validation set, which in this case is more commonly referred to as test set in the ML community. So was the test set used for hyperparameters tuning? In this case I would suggest to call it validation set, and show also the results of the hyperparameter tuning.

4. The qualitative analysis of sensitivity and specificity is nice, but it should be shown more systematically. A table of precision and recall, or the F1 score, should be provided.

5. Since the decrease in AP for near objects is unexpected, at least some justification for it should be provided.

6. The final part about fallen trees is too incomplete to be shown, it should be left for future work.

7. The quality of the English language has to be improved. In particular the use of the articles "a" and "the" should be considered.

Reviewer 2 Report

-The paper should be interesting ;

-it is a good idea to add more photos of measurements, sensors + arrows/labels what is what  (if any);

-What is the result of the analysis?;

-figures should have high quality, TIFF format.

-text should be formatted;

-labels of figures should be bigger;

-please add photos of the application of the proposed research, 2-3 photos ;

-what will society have from the paper?;

-is there a possibility to use the proposed approach for other problems?

-please compare advantages/disadvantages of other approaches etc.;

-references should be from the web of science 2020-2022 (50% of all references, 30 references at least);

-please compare advantages/disadvantages other approaches;

-Conclusion: point out what have you done;

Round 2

Reviewer 1 Report

The authors have addressed most of my comments. The only remaining ones are:

- the dataset definition is still messy and unclear. It is once mentioned that the real-world images are 804, and then they are 1005. And moreover, the authors didn't answer the question on how were the images selected. Randomly?

- for the next time, please highlight in the manuscript the parts that were changed with respect to the previous version, so as to help the reviewers in their work.

Author Response

Point 1: the dataset definition is still messy and unclear. It is once mentioned that the real-world images are 804, and then they are 1005. And moreover, the authors didn't answer the question on how were the images selected. Randomly?

Response 1: In order to improve the clarity on dataset definition, the related paragraph of Section 4-Evaluation has been thoroughly re-written. A sentence is added on how images were selected. In addition, a table with numbers defining the related datasets sizes has been included.

Point 2: for the next time, please highlight in the manuscript the parts that were changed with respect to the previous version, so as to help the reviewers in their work.

Response 2: We thank the reviewer for pointing out this mistake. This time we upload the manuscript in tracking mode to visualize the changes we have done.

Reviewer 2 Report

-arrows + labels to figures should be added, Fig. 3, Fig. 4, Fig. 6

Author Response

Point 1: arrows + labels to figures should be added, Fig. 3, Fig. 4, Fig. 6. 

Response 1: Figures 3, 4, 5, 6 have been updated by adding arrows+labels+figures. Captions are updated, as appropriate.